

# Irreversible ocean thermal expansion under negative $CO_2$ emissions

Dana Ehlert[1] and Kirsten Zickfeld[1]

[1]Department of Geography, Simon Fraser University, 8888 University Drive, Burnaby, B.C., Canada V5A 1S6

*Correspondence to:* Dana Ehlert (dehlert@sfu.ca)

**Abstract.** In the Paris Agreement in 2015 countries agreed on holding global mean surface air warming well below 2°C but the emission reduction pledges under that agreement are not ambitious enough to meet this target. Therefore, the question arises whether restoring temperature to this target after exceeding it by artificially removing $CO_2$ from the atmosphere is possible. One important aspect regards the reversibility of ocean heat uptake and the associated sea level rise, which have very long

(centennial to millennial) response time scales. In this study the response of sea level rise due to thermal expansion (TSLR) to a 1% yearly increase of atmospheric $CO_2$ up to a quadrupling of the pre-industrial concentration followed by a 1% yearly decline back to the pre-industrial $CO_2$ concentration is examined using the University of Victoria Earth System Climate Model (UVic ESCM). We find that TSLR continues for several decades after atmospheric $CO_2$ starts to decline and that sea level does not return to pre-industrial levels for over thousand years after atmospheric $CO_2$ is restored to pre-industrial concentrations.

This finding is independent of the strength of vertical sub-grid scale ocean mixing implemented in the model. Furthermore, TSLR rises faster than it declines in response to a symmetric rise and decline in atmospheric $CO_2$ concentration partly because the deep ocean continues to warm for centuries after atmospheric $CO_2$ returns to the pre-industrial concentration. Both TSLR rise and decline rate increase with increasing vertical ocean mixing. Exceptions from this behaviour arise if the overturning circulations in the North Atlantic and Southern Ocean intensify beyond pre-industrial levels in model versions with lower

vertical mixing, which leads to rapid cooling of the deep ocean.

## 1 Introduction

Policy makers agreed to "Holding the increase in the global average temperature to well below 2°C above pre-industrial levels and pursuing efforts to limit the temperature increase to 1.5°C above pre-industrial levels" under the Paris Agreement in 2015 (Paris Agreement, 2015). But the national pledges under this agreement are not sufficient to meet this target (Peters et al.,

2015; Rogelj et al., 2016). At the same time, global mean temperature would remain elevated even if all $CO_2$ emissions were to cease (Matthews and Caldeira, 2008; Gillett et al., 2011; Matthews and Zickfeld, 2012; Zickfeld et al., 2013), implying that the mere cessation of emissions would not enable recovery of a warming target after exceeding it. Therefore, the idea has been discussed of artificially removing $CO_2$ from the atmosphere, a measure referred to as "negative $CO_2$ emissions", with a number of techniques such as reforestation or direct air capture (Smith et al., 2016). It should be noted though that none of

these techniques has yet been applied on a large scale (Fuss et al., 2016; Smith et al., 2016). However, most future scenarios not exceeding the 2°C warming target include such negative $CO_2$ emissions and show a peak and decline in atmospheric $CO_2$



stronger than would be observed from only zeroing emissions (Smith et al., 2016). For example, the only Representative Concentration Pathway not exceeding 2°C, RCP 2.6, includes negative $CO_2$ emissions (Meinshausen et al., 2011). The decline in radiative forcing induced by negative $CO_2$ emissions rises the question of the reversibility of anthropogenic climate change, i.e., to what extent it is possible to revert to either a pre-industrial climate or another warming target such as 2°C and the related

climate state by artificially removing $CO_2$ from the atmosphere.

Previous studies have shown that global surface air warming is reversible on human time scales but lags the $CO_2$ decline due to the ocean's thermal inertia (Boucher et al., 2012; Tokarska and Zickfeld, 2015; Zickfeld et al., 2016). Other aspects, such as precipitation or sea ice, also decline but lag the temperature response (Boucher et al., 2012). The effectiveness of negative $CO_2$ emissions to lower atmospheric $CO_2$ is impeded as $CO_2$ outgases from natural carbon sinks in reaction to the negative

$CO_2$ emissions. This outgassing increases with increasing negative emissions, thus negative emissions become less effective the higher they get (Tokarska and Zickfeld, 2015). The total negative emissions for reverting to the initial $CO_2$ concentration are higher than the total positive emissions if the permafrost carbon feedback, which accounts for additional carbon emissions from thawing permafrost, is included (MacDougall, 2013). This means that more carbon needs to be artificially removed from the atmosphere than was initially emitted due to the hysteresis behaviour of the permafrost carbon pool. Another important

finding of the reversibility research under negative emissions is that the approximately constant ratio between temperature change and cumulative emissions differs between atmospheric $CO_2$ increase and decline phases. This difference would need to be taken into account when setting total allowable emissions for a certain warming target after overshoot (Zickfeld et al., 2016).

Previous studies found thermosteric sea level rise to be in principle reversible (Bouttes et al., 2013; Zickfeld et al., 2013;

MacDougall, 2013), but reversing it on centennial time scales requires large amounts of negative $CO_2$ emissions, which are likely infeasible with currently discussed technologies (Tokarska and Zickfeld, 2015). By using a 2-layer ocean model (Gregory, 2000; Geoffroy et al., 2013) Bouttes et al. (2013) show that the decline in thermosteric sea level in response to zeroed or negative radiative forcing (with preceding positive radiative forcing) is due to a strong temperature decline in the upper ocean relative to the deep ocean, which enables the release of heat. This decline in thermosteric sea level cannot be explained

with a zero dimensional energy balance model (0-D EBM) where the ocean is modelled as an infinite heat sink, which cannot spontaneously release heat. Limitations of the 2-layer model are that is has only two different heat capacities and it does not include ocean overturning circulations. Both shortcomings lead to a too slow heat release under negative emissions relative to an Atmosphere Ocean General Circulation Model (Fast Met Office/UK Universities Simulator) (Bouttes et al., 2013).

Zickfeld et al. (2017) investigate the mechanisms of thermosteric sea level rise and decline induced by emissions of short-lived

radiative forcing agents, such as methane, and the cessation of those emissions. The thereby induced increase and decline in radiative forcing is similar to the change in radiative forcing that can be induced by positive and negative $CO_2$ emissions as short-lived forcing agents decline relatively fast in the atmosphere after cessation of their emissions. Zickfeld et al. (2017) show that the rate of sea level change from thermal expansion can be approximated with the difference between radiative forcing and a term representing radiative damping to space, which corresponds to a 0-D EBM (the difference relative to the 0-D EBM

used by Bouttes et al. (2013) is that they do not assume ocean heat uptake to be proportional to temperature change, which



is justified for increasing but not decreasing radiative forcing). This model shows that a negative difference between radiative forcing and global mean surface air temperature (GMSAT) change scaled by the climate feedback parameter enables declining global mean ocean temperatures and thus declining thermosteric sea level.

Herein, we further explore the physical mechanisms that determine the reversibility of thermosteric sea level rise. In particu-

lar, we examine the sensitivity of the reversibility of thermal expansion to parameterization of sub-grid ocean mixing. This is important because parameterization of sub-grid ocean mixing is still an uncertainty in climate models that strongly affects the timescale of ocean heat uptake and release. Using a range of ocean mixing parameters also enables further insights into the processes involved in the reversibility of thermal expansion and the associated sea level change. We examine the reversibility of thermal expansion and the processes involved in the context of symmetrically increasing and decreasing $CO_2$ concentrations.

The model used for this study is a model of intermediate complexity, which allows for simulations on long (millennial) time scales. Section 2 describes this model and the simulations performed. The results of these simulations are presented in Section 3. In Section 4 the conclusions from those results are drawn and discussed.

## 2   Model and Simulations

### 2.1   Model

Simulations for this study were performed using the University of Victoria Earth System Climate Model version 2.9 (UVic ESCM 2.9) (Eby et al., 2009), a model of intermediate complexity. The physical model consists of an atmosphere energy balance model, a general circulation ocean model, a sea ice model, and a land surface scheme. All components include coupled carbon cycle descriptions and have a resolution of 1.8°(meridional) x 3.6°(zonal). The model does not include an ice sheet model and we only discuss the sea level rise due to thermal expansion of the ocean in the following sections.

The atmosphere model is a vertically integrated energy-moisture balance model, which includes wind, planetary long wave, and water vapour feedbacks. Clouds are represented in the albedo of the atmosphere but cloud feedbacks are not included. The land is represented using a simplified version of the Met Office Surface Exchange Scheme (MOSES) (Meissner et al., 2003; Cox et al., 1999), coupled to the dynamic vegetation model TRIFFID (Top-down Representation of Interactive Foliage and Flora Including Dynamics) (Cox, 2001).

The ocean is described with a three dimensional general circulation model with 19 vertical layers. It is the Geophysical Fluid Dynamic Laboratory (GFDL) Modular Ocean Model (MOM) (Weaver et al., 2001), which is coupled to a thermodynamic sea ice model, an inorganic carbon cycle model, a marine biology model (Schmittner et al., 2005) and a sediment model. Sub-grid ocean mixing is described via momentum diffusivity (or viscosity) and tracer diffusivity (Weaver et al., 2001). The following discussion focuses on parametrization of tracer diffusivity and the term ocean mixing only refers to the diffusion of tracers.

This diffusion of tracers is parameterized as diffusion along isopycnals (surfaces of constant density) and diffusion across isopycnals. The diffusivity along isopycnals is set to 800 $m^2 s^{-1}$ and an additional parametrization is implemented to account for instabilities where isopycnals are tilted (Gent and Mcwilliams, 1990). This additional parameter, referred to as Gent & McWilliams thickness diffusivity, is also set to 800 $m^2 s^{-1}$. Mixing across isopycnals (diapycnal mixing) is described via a ver-





tical mixing scheme as there is no practical difference between vertical and diapycnal mixing due to isopycnal slope limitations in the ocean model. There are three vertical mixing scheme options in the UVic ESCM 2.9: a vertically and laterally constant mixing scheme, a depth-dependent but laterally constant (Bryan & Lewis) (Bryan and Lewis, 1979) mixing scheme, and a tidal mixing scheme, where mixing due to the dissipation of tidal energy over topography is added to a constant background
diffusion parameter (Schmittner et al., 2005).

## 2.2  Simulations

The UVic ESCM is spun up for 6000 years under pre-industrial (year 1800) conditions for different model versions using different vertical ocean mixing parameter values and schemes (Table 1). The range of mixing parameters is chosen to achieve the widest possible range in ocean heat uptake while keeping the model stable and not necessarily to use parameters that closely
reproduce observed ocean tracer distributions. However, for the tidal mixing scheme the range of background diffusivities was informed by studies that aimed at finding a range that best fits tracer observations (Schmittner et al., 2009; Goes et al., 2010; Ross et al., 2012). Mixing along isopycnals only entails small variations in ocean heat uptake (Ehlert et al., 2017) and is therefore set to the default value in all model versions.

The default setting has a Bryan & Lewis vertical mixing scheme and a diffusivity of 0.3 $cm^2s^{-1}$ in the deeper ocean and 1.3
$cm^2s^{-1}$ in the upper ocean. This range is shifted to higher values of 0.5-1.5 $cm^2s^{-1}$ ($k_{v,B\&L}$high) and lower values of 0.1-1.1 $cm^2s^{-1}$ ($k_{v,B\&L}$low) while the vertical distribution within these ranges remains the same. The mixing scheme was changed from Bryan & Lewis to a vertically constant mixing ($k_{v,const}$), where the diffusivity was set to a value between 0.05 $cm^2s^{-1}$ and 1.0 $cm^2s^{-1}$. Additionally, a tidal mixing scheme is used ($k_{v,tidal}$) and the background diffusivity was set to a value between 0.1 $cm^2s^{-1}$ and 0.45 $cm^2s^{-1}$.
All model versions are run to a pre-industrial equilibrium and subsequently forced with an idealized scenario of a 1% yearly increase in atmospheric $CO_2$ concentration until quadrupling of the pre-industrial concentration (simulation year 140) followed by a 1% yearly decrease until pre-industrial levels are reached (simulation year 280). The simulations are continued with constant pre-industrial $CO_2$ concentration for another 1120 simulation years.

## 25  3  Results

### 3.1  Reversibility of sea level rise

In this section we focus on the discussion of thermosteric sea level rise and decline for the default ocean mixing setting (black curves in all figures with line plots) and Section 3.2 discusses the effect of different ocean mixing settings on this reversibility. Global mean surface air temperature (GMSAT) declines shortly after a decline in $CO_2$ is prescribed (Figure 1a). Global mean
thermosteric sea level (GMTSL) continues to rise for another 80 years until it starts declining (Figure 1b). The decline is much slower than the rise in GMTSL despite a symmetrically increasing and decreasing atmospheric $CO_2$ concentration. To further





**Table 1.** Description of different model versions and their names as referred to in the text and figures. $k_v$ is the vertical diffusivity. The pre-industrial state of the different model versions is shown exemplarily for the following variables: global mean surface air temperature (SAT), Atlantic Meridional Overturning Circulation (AMOC) and the given values refer to the maximum of the overturning stream function.

| Experiment | vertical mixing scheme | $k_v$ ( $cm^2s^{-1}$ ) | SAT (°C) | AMOC (Sv) |
|---|---|---|---|---|
| default | Bryan & Lewis | 0.3-1.3 | 13.39 | 21.6 |
| $k_{v,B\&L}$low | Bryan & Lewis | 0.1-1.1 | 13.16 | 12.7 |
| $k_{v,B\&L}$high | Bryan & Lewis | 0.5-1.5 | 13.52 | 25.8 |
| $k_{v,const}$0.05 | vertically constant | 0.05 | 12.89 | 9.8 |
| $k_{v,const}$0.3 | vertically constant | 0.3 | 13.33 | 21.4 |
| $k_{v,const}$1.0 | vertically constant | 1.0 | 16.67 | 32.3 |
| $k_{v,tidal}$0.1 | tidal | 0.1 | 13.15 | 14.4 |
| $k_{v,tidal}$0.2 | tidal | 0.2 | 13.32 | 19.3 |
| $k_{v,tidal}$0.45 | tidal | 0.45 | 13.49 | 25.8 |

investigate the mechanism of the GMTSL rise and decline we investigate ocean temperature as a proxy for GMTSL as the depth profile of ocean temperature gives insight into the distribution of heat in the ocean and will enable comparison with a simple ocean model in Section 3.2. The choice of ocean temperature as a proxy for thermosteric sea level rise is reasonable as changes in global mean ocean temperature (GMOT) and changes in GMTSL are nearly linearly related and follow a similar

temporal evolution (Figure 1b,c).

The decline in GMOT and thus GMTSL is very slow because the warming signal from the previous increase in $CO_2$ still penetrates into the deeper ocean centuries after the radiative forcing has returned to zero and heat that entered the mid ocean is mixed into the upper ocean very slowly (Figure 2a). Thus, despite a cooling of the upper ocean, the deep ocean is still heating, except at high latitudes where the deep ocean cools (Figure 3b). This cooling is likely associated with an intensification of

deep and bottom water formation, which overshoots the original value after $CO_2$ concentrations return to pre-industrial levels (Figures 6 and 7; these responses will be discussed in further detail in Section 3.2). The decline in GMOT lags the decline in atmospheric $CO_2$ concentration and also the decline in global mean surface air temperature (GMSAT) to such an extent that by the end of the simulation (1120 years after $CO_2$ concentration is restored to pre-industrial levels) GMOT and thus thermosteric sea level have not returned to pre-industrial levels. As seen in previous studies (Boucher et al., 2012), GMOT exhibits hysteresis

relative to change in radiative forcing as GMOT remains elevated despite zero radiative forcing (Figure 1d). This hysteresis behaviour is due to a lagged response of GMOT to $CO_2$, rather than a change in the state of the ocean.

Previously it has been assumed that GMTSL change is proportional to GMSAT changes and thus to radiative forcing (Rahmstorf, 2007; Bouttes et al., 2013). However, this assumption does not hold for declining radiative forcing (Bouttes et al., 2013) and Zickfeld et al. (2017) therefore suggested the following relationship, which holds also under declining radiative forcing:

$$\frac{d\eta}{dt} = \alpha RF - \beta \Delta T$$







**Figure 1.** Changes in global mean surface air temperature (a), global mean thermosteric sea level rise (b), and global mean ocean temperature (c) relative to year 0 over time and change in GMOT versus radiative forcing. The first vertical grey line in panel a, b, and c indicates the time of peak atmospheric $CO_2$ concentration (quadrupling of pre-industrial levels, year 140). The second grey line in panel b and c indicates the time GMTSL and GMOT reach maximum values. The third grey line in panel c is the time Figures 2 and 3 refer to year 1100.

where $\eta$ is the sea level rise due to thermal expansion, RF is the radiative forcing, and $\Delta T$ is the temperature change relative to a reference year. $\beta/\alpha$ is equal to the climate feedback parameter $\lambda$ in $Wm^{-2}K^{-1}$ by analogy of the above equation with the zero dimensional energy balance equation (i.e., $d\eta/dt \propto RF-\lambda\Delta T$). Applying this framework to the simulation results discussed here shows that while GMTSL is rising (i.e., positive time derivative) the radiative forcing is larger than $\lambda\Delta T$ (Figure 4). However,




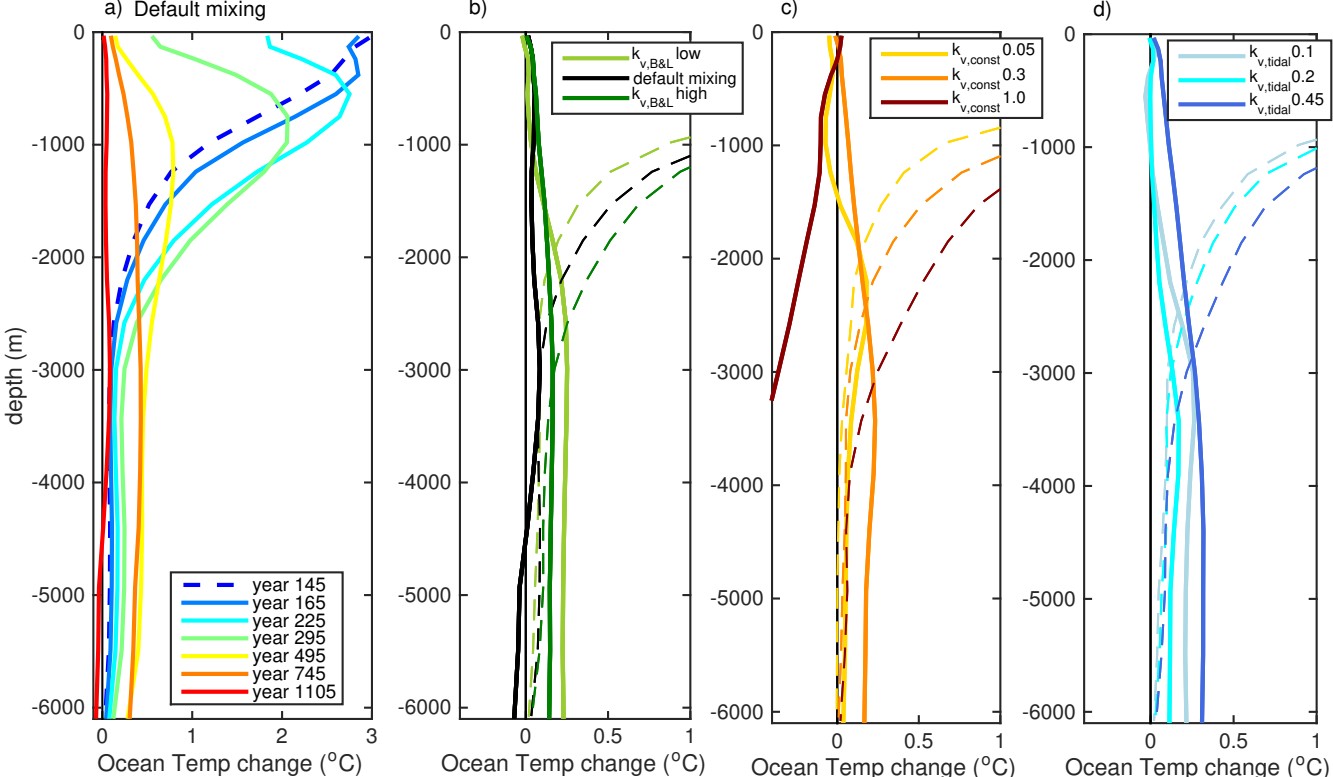

**Figure 2.** Zonal and meridional average of change in ocean temperature relative to year 0 for the model version with default mixing setting for various points in time (a). Zonal and meridional average of change in ocean temperature relative to year 0 for model versions with different ocean mixing settings for year 140 (dashed curves, year of peak forcing) and year 1100 (continuous curves, 3rd vertical grey line in figure 1c) for Bryan& Lewis mixing scheme (b), constant mixing scheme (c), and tidal mixing scheme (d).

when $\lambda \Delta T$ is larger than the radiative forcing, induced by the lag of the GMSAT decline relative to the decline in radiative forcing, GMTSL declines. Thus the decline of GMOT and GMTSL is due to a negative radiation imbalance at the top of the atmosphere.

## 3.2 The effect of ocean mixing on sea level rise and its reversibility

5 In this section we discuss the effect of different parameters for vertical ocean mixing on GMTSL rise and decline and identify the mechanisms that lead to the differences in GMTSL among model versions with different vertical mixing parameters. Under increasing atmospheric $CO_2$ concentration, GMTSL rises faster with increasing vertical mixing parameter (Figure 1b,c). Peak GMTSL, which occurs around year 220 (80 years after atmospheric $CO_2$ started declining), ranges between 0.45 m and 0.80 m. Similarly to the increase, GMTSL and GMOT also decrease faster under increased vertical ocean mixing, with

10 exceptions for $k_{v,B\&L}$high and $k_{v,tidal}$0.45 simulations (Figure 1b,c). In these simulations, GMOT decrease rate is similar to



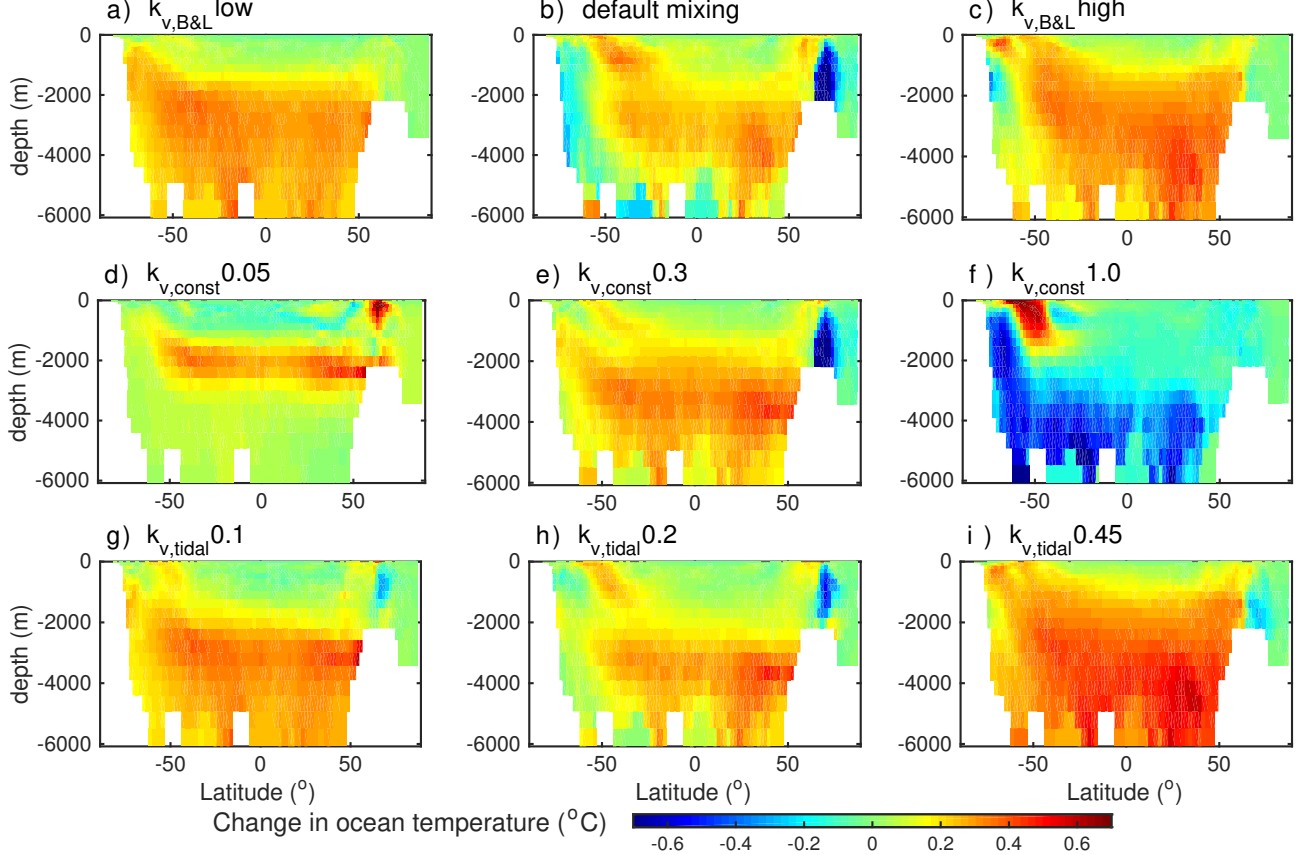

**Figure 3.** Zonal average change in ocean temperature for year 1100 relative to year 0 for the different model versions. Each row shows results for one mixing scheme (Bryan&Lewis mixing, constant mixing, tidal mixing in the 1st, 2nd, and 3rd row, respectively) and the vertical diffusivity increases from left to right.

the rate in model versions with lower mixing parameters. The causes for this behaviour will be discussed later in this section. In most cases a higher decline rate of GMOT for a model version with higher vertical diffusivity results in a crossover of the GMOT curves (Figure 1b,c): for example, the $k_{v,B\&L}$ low simulation has a slower ocean warming than the default simulation between year 0 and year 900. However, the rate of ocean temperature decline is higher in the default simulation than in the

5  $k_{v,B\&L}$ low simulation, leading to a crossover of these two ocean temperature curves, and from around year 900 on the default simulation has lower ocean temperatures and thus a lower GMTSL.

This behaviour of faster GMOT increase and decline rates under higher mixing, including the crossover of the GMOT curves, can be explained with a 2-layer ocean model (Figure 5a). This model consists of an upper ocean layer and a deep ocean layer (Gregory, 2000; Bouttes et al., 2013). The upper layer is thin (depth $d_u$ 100 m) and responds immediately to changes in forcing because it has a small heat capacity. The lower layer is thick (depth $d_l$ 2000 m), has a high heat capacity, and thus provides

10  the inertia of the ocean response. The heat flux between the upper and lower layer is proportional to the temperature difference




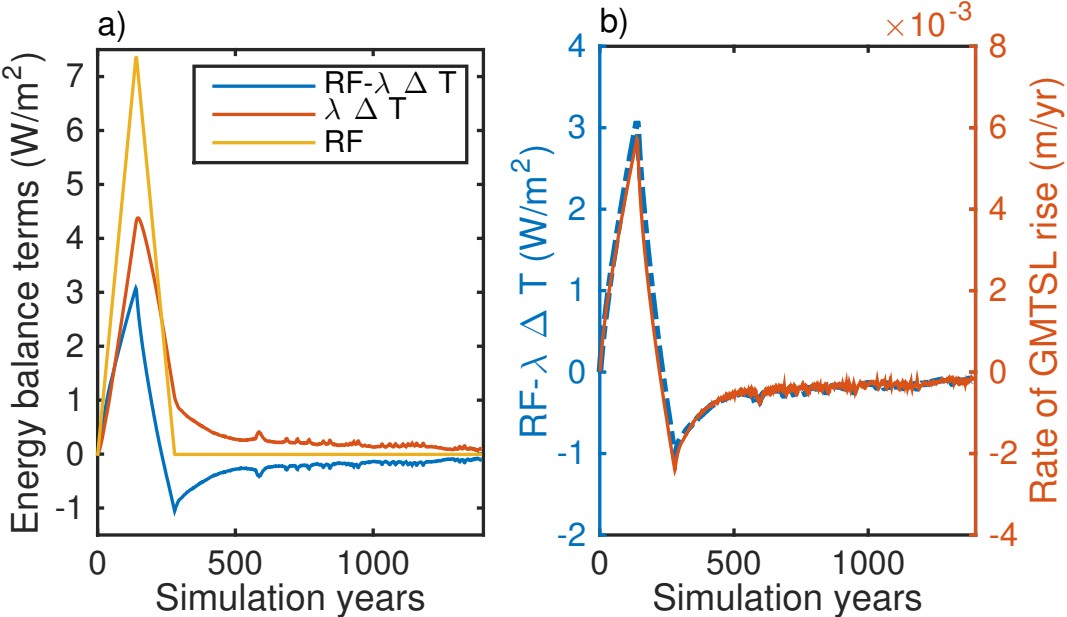

**Figure 4.** Energy balance terms calculated from the simulation data of the model version with default mixing setting. $\beta/\alpha = \lambda = 1$ $Wm^{-2}K^{-1}$ is the climate feedback parameter for the default mixing setting. $\Delta T$ is the change in global mean surface air temperature relative to year 0 and RF is the radiaitve forcing. Panel (b) compares the energy imbalance (blue y-axes and curve) to the rate of Global Mean Thermosteric Sea Level rise (red y-axes and curve).

between these layers. The temperature for each layer ($T_u$ temperature upper layer, $T_l$ temperature lower layer) can be calculated using the following equations:

$$cd_u\frac{dT_u}{dt} = RF - ck\frac{(T_u - T_l)}{0.5(d_l + d_u)} - \lambda T_u$$

$$cd_l\frac{dT_l}{dt} = ck\frac{(T_u - T_l)}{0.5(d_l + d_u)}$$

5  where c= $4.218*10^6$ $Jm^{-3}K^{-1}$ is the volumetric heat capacity, k =$1*10^{-4}m^2s^{-1}$= 1 $cm^2s^{-1}$ is the thermal diffusivity between the layers, $\lambda$= 1.0 $Wm^{-2}K^{-1}$ is the climate feedback parameter, and RF ($Wm^{-2}$) is the surface heat flux representing the external radiative forcing, which is prescribed according to a symmetric 1% yearly increase to quadrupling of pre-industrial $CO_2$ levels and subsequent decrease in atmospheric $CO_2$, i.e., the same forcing as in the UVic ESCM simulations. Changes in mixing are achieved by changing the diffusivity k between the layers in the range from 0.05*k to 1.0*k. This range was chosen

10  to correspond to the range in mixing parameters in the UVic ESCM simulations ($k_{v,const}$0.05 and $k_{v,const}$1.0) that leads to the widest range in GMOT.

 The temperature of the upper layer reacts instantaneously to changes in the forcing (Figure 5a). The temperature in the lower layer lags behind the radiative forcing and dominates the total temperature of the ocean (Figure 5b,d) due to its very high heat capacity relative to the heat capacity of the upper layer. Therefore, ocean heat uptake or release can be approximated by the





**Figure 5.** Temperature changes for the 2-layer ocean model: change in upper ($T_{upper}$) (a) and lower layer ocean temperature $T_{lower}$ (b), difference in the temperature between the layers (c) and change in total ocean temperature (d, calculated as average from $T_{upper}$ and $T_{lower}$ weighted by the layer depth). Change in total ocean temperature versus radiative forcing (e). k=1 cm$^2$s$^{-1}$ is the thermal diffusivity between the layers.

heat exchange between the layers, which is determined by the thermal diffusivity and the temperature gradient between the layers. A higher diffusivity between the layers enables a faster heat exchange between the layers and a faster heat uptake and release by the whole ocean as observed in the model (Figure 5d). However, the temperature gradient between the layers is





lowest for the model version with the highest diffusivity (Figure 5c) implying a diminished heat exchange between the layers and a slower ocean heat uptake and release, which is the opposite from what is observed in the model. Therefore, it is the effect of the diffusion on ocean heat uptake that dominates over the effect of the vertical temperature gradient. This 2-layer model also shows a similar hysteresis behaviour as the UVic ESCM simulation (compare Figures 1d and 5e).

One limitation of the two-layer model is that the heat capacity of the lower layer is too high, which results in a very slow heat release from this layer. This effect is especially strong for very low diffusivity values (Figure 5b blue and red curve). For comparison: in a multi-layer model where heat is transferred step by step from one layer to the next, heat is more easily released because upper layers, which extend deeper than the upper layer in the 2-layer model, have a response time scale that is between that of the upper and lower layer of the 2-layer model and therefore release heat faster than the lower layer in the 2-layer

model (Bouttes et al., 2013). The other limitation is that the 2-layer model does not include a representation of the meridional overturning circulation, which intensifies in the UVic ESCM simulations in both the Atlantic Ocean and the Southern Ocean after radiative forcing returned to zero and leads to a stronger heat release. Due to these two shortcomings the decline in ocean temperature is much slower in the 2-layer model than in the UVic ESCM and 2-layer model simulations have to be extended for 1000 years compared to the UVic ESCM simulations to observe a crossover of the ocean temperature curves.

Generally, the initial (pre-industrial) meridional overturning in both the Atlantic Ocean and the Southern Ocean increases as diffusivity increases (Figure 6a and 7a) in the UVic ESCM model versions. In addition, response of the meridional overturning circulations to the radiative forcing also differs among model versions. These differences in the overturning response likely lead to divergences between the results from UVic ESCM simulations and 2-layer ocean model simulations as will be discussed in the following.

Cases where the UVic ESCM simulations diverge from the 2-layer ocean model behaviour (i.e., similar GMOT decline rate in model version with higher and lower diffusivity) are the model version with default ocean mixing setting relative to the $k_{v,B\&L}$ high model version and the $k_{v,tidal}0.2$ relative to the $k_{v,tidal}0.45$ model version. For each model version pair, the uptake of heat is faster for the model version with the higher mixing but the rate of heat release is approximately the same. This similar ocean heat release rate is likely caused by a stronger intensification of the overturning in both the Atlantic and the

Southern Ocean in response to the declining radiative forcing in the simulations with lower vertical mixing (i.e., default mixing setting and $k_{v,tidal}0.2$).

The globally averaged vertical ocean temperature profiles illustrate the finding that higher mixing does not always entail a stronger decline in ocean temperatures (Figure 2b,d). At year 140 (year of peak forcing), for the Bryan& Lewis mixing scheme (default mixing setting, $k_{vB\&L}$ low and high) and the tidal mixing scheme ($k_{v,tidal}$), the temperature in the surface and mid-

ocean increases with increasing ocean mixing. However, for year 1100 default mixing and $k_{v,tidal}0.2$ simulations have the lowest ocean temperature below a depth of around 1100 m. Interestingly, in those cases the deep ocean has also cooled more than the mid ocean, probably induced by increased Antarctic Bottom Water (AABW) formation. For the tidal mixing scheme, both simulations with the lower mixing setting ($k_{v,tidal}0.1$ and 0.2) have a stronger cooling than the simulation $k_{v,tidal}0.45$. The discussion of the overturning circulation and especially its response to the declining radiative forcing will give further

insights into this behaviour.




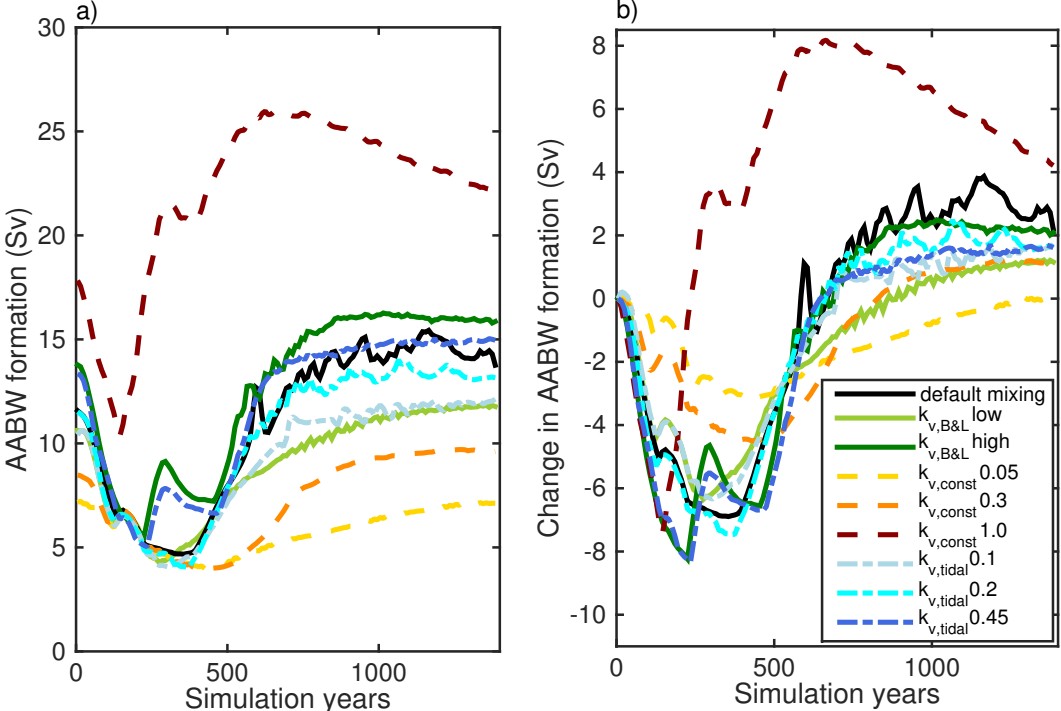

**Figure 6.** a) Antarctic Bottom Water (AABW) formation, calculated as minimum of the zonally averaged overturning stream function (averaged over the globe) below 500 m. b) Change in the AABW formation relative to year 0.

Both Atlantic Meridional Overturning Circulation (AMOC) and AABW formation decline in response to increased atmospheric $CO_2$ concentrations and the associated global warming (Figures 6a, 7b). AMOC responses to global warming have been well studied (Stocker and Schmittner, 1997; Rahmstorf, 2006; Meehl et al., 2007) and the decline in the AMOC under warming is due to surface warming and freshening of North Atlantic surface waters (Rahmstorf, 2006). The AMOC increases

again in a delayed response to the decrease in radiative forcing (Figure 7a). This decrease leads to a cooling of surface waters and also increase in sea ice formation that increases surface water density and thus increases the deep water formation. The AMOC even overshoots its pre-industrial strength (Figure 7b) due to a build up of salinity in the subtropical gyre in the northern Atlantic while AMOC is weaker (Wu et al., 2011). When the AMOC strengthens again as a response to declining surface temperature this salinity anomaly is advected northward, which results into denser water in the North Atlantic and an

intensified AMOC. This AMOC overshoot has been linked to an increased rate of ocean heat release (Bouttes et al., 2015) and thus a stronger GMTSL decline. Earlier studies (Knutti and Stocker, 2000; Levermann et al., 2005) found a slow increase (over several millennia) of global mean sea level after a shut down of the AMOC because the reduction of the surface temperature induced by the AMOC shutdown reduces the radiation lost to space and thereby increases ocean heat uptake. A rapid regional thermosteric sea level increase in the North Atlantic with thermosteric sea level decrease in the Southern Ocean is also shown

by Levermann et al. (2005). This slow increase of global sea level is in contrast to the link we propose whereby GMTSL rises



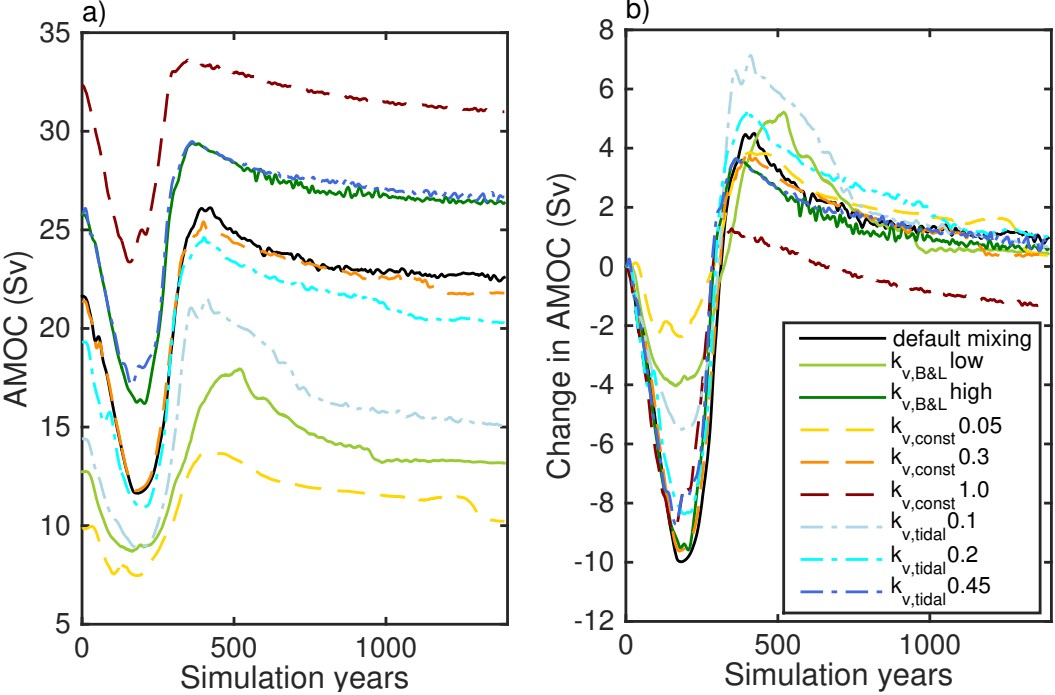

**Figure 7.** a) Atlantic meridional overturning circulation (AMOC) calculated as maximum of the zonally averaged overturning stream function in the North Atlantic. b) Change in the AMOC relative to year 0.

less with a weaker AMOC. However, the mechanism between a slowdown of the AMOC and a complete shutdown could be different because a slightly weaker AMOC does not induce such strong changes in surface temperature. Declining AABW formation/volume in response to anthropogenic climate change has been seen in models and observational data (Purkey and Johnson, 2012) but the causes for this response have not been explored to our knowledge. The AABW formation increases as a

delayed response to the decreasing radiative forcing (Figure 6a), likely due to surface cooling and increased sea ice formation, and it also intensifies above pre-industrial levels (Figure 6b). However, this intensification is much smaller and slower than the AMOC overshoot. Further investigation is needed to uncover the processes that cause this intensification.

Both AMOC and AABW formation overshoot/intensify more strongly in the model versions with lower mixing setting (i.e., in the model version with default mixing relative to $k_{v,B\&L}$high, and $k_{v,tidal}$0.1 and 0.2 relative to $k_{v,tidal}$0.45), which leads to

faster heat release as the exchange between deep/bottom waters and surface waters is enhanced. This intensification in overturning circulation likely offsets the slower heat exchange between upper and deeper ocean in the model version with lower mixing (i.e., default mixing setting and $k_{v,tidal}$0.2). The cooling effect on the deep and bottom waters from stronger than pre-industrial AMOC and AABW formation is also evident in the zonally averaged ocean temperature changes (Figure 3), where the simulations with default ocean mixing and $k_{v,tidal}$0.2 have more cooling in the mid ocean at northern high latitudes and

the deep ocean at southern high latitudes than in the $k_{v,B\&L}$high and $k_{v,tidal}$0.45 simulations. The deep ocean cooling is less





pronounced in the $k_{v,tidal}$0.2 case. Also the $k_{v,tidal}$0.1 has a very strong AMOC overshoot, which together with the lower heat uptake during the increasing forcing phase results in a stronger cooling than in the $k_{v,tidal}$0.45 simulation.

A special case is the $k_{v,const}$1.0 simulation, which has the highest vertical mixing of all simulations. Here, the GMOT and thus GMTSL decline below pre-industrial levels implying that more heat is being released than initially entered the ocean. This

strong cooling is likely linked to a strong intensification of the AABW formation under this very high mixing setting (Figure 6b, red curve). This assumption is supported by a strong cooling in the Southern Ocean along the path of AABW (Figure 3f). The intensification of AABW formation is likely enabled by a deep convection cell that persists in this model version for multiple centuries in the Southern Ocean. Due to the high vertical mixing in this model version, the deeper ocean is relatively warm (compared to other model versions), which causes instabilities in the Southern Ocean where surface water are very cold

and convection cells form. While atmospheric $CO_2$ increases these cells only persist for a decade or less and therefore do not have a significant effect on ocean heat uptake and GMTSL. However, when atmospheric $CO_2$ and thus GMSAT decline the surface ocean cools, which reinforces the deep convection and enables for the convection cell to persist for centuries. High vertical mixing at all depth is not observed and therefore this model version and the strong decline in GMTSL may not be realistic.

**4    Discussion and Conclusions**

In this study the reversibility of thermosteric sea level rise in response to a symmetric increase and decrease in atmospheric $CO_2$ concentration is examined. Furthermore, the sensitivity of this reversibility to sub-grid scale ocean mixing is investigated, which also gives further insight into the role of ocean circulation in this reversibility. Different versions of the UVic ESCM 2.9, which differ in the parametrization of vertical sub-grid scale ocean mixing, are forced with a 1% yearly increase in atmo-

spheric $CO_2$ concentration until quadrupling of pre-industrial levels, followed by a 1% yearly decrease in $CO_2$ and constant pre-industrial $CO_2$ concentration thereafter. Such a strong decline in atmospheric $CO_2$ concentrations can only be realised with net negative $CO_2$ emissions, i.e., artificial removal of $CO_2$ from the atmosphere. The net negative $CO_2$ emissions required to achieve a 1% yearly decline in atmospheric $CO_2$ are likely unfeasible with current technologies (Boucher et al., 2012; Tokarska and Zickfeld, 2015) and the results shown in this study are therefore conservative.

We find that thermosteric sea level is not reversible on human time scales (decadal to centennial time scales) as GMTSL rise continues for 80 years after atmospheric $CO_2$ concentration starts declining and drops only slowly thereafter. The decline of GMTSL is much slower than the rise due to the thermal inertia of the ocean, which is expressed by a warming signal in the deeper ocean that persists for almost a millennium after atmospheric $CO_2$ concentration is restored to pre-industrial levels. Furthermore, GMTSL does not revert to pre-industrial values by the end of the simulations (1120 years after atmospheric $CO_2$

concentration is restored to pre-industrial levels). The release of heat by the ocean, and thus a declining GMTSL, are explained by invoking a 0-D EBM where the rate of GMTSL rise is linked to radiative forcing and a term representing radiative damping to space. This model shows that a declining GMTSL is enabled by radiative forcing being lower than the change in GMSAT scaled by the climate feedback parameter.





Generally, sea level rise and decline rates in response to increasing and decreasing atmospheric $CO_2$ increase with higher vertical diffusivity, which can be explained with a simple 2-layer ocean model with diffusion of heat between a thin upper layer and a deep lower layer. Exceptions to this behaviour in the UVic ESCM simulations are linked to a strengthening of the meridional overturning circulation beyond pre-industrial values (inferred from North Atlantic deep water, NADW, and AABW formation

intensification) after recovering from a decline induced by the increasing radiative forcing. Generally, this intensification in meridional overturning circulation beyond pre-industrial values is stronger in model versions with lower vertical diffusivity. In some cases this intensification is so large in the model version with lower diffusivity that it offsets the effect of decreased diffusivity and GMTSL decline rates are similar among model versions with different diffusivity values. Stronger NADW and AAWB formation increase the exchange of heat between the upper and the deeper ocean and the intensification of this deep

and bottom water formation beyond their pre-industrial levels increases the release of heat from the ocean. Especially the intensification of AABW formation results in a cooling of the deep ocean after radiative forcing has returned to zero.

Limitations of the research presented here are that the model used (UVic ESCM 2.9) does not include an interactive ice sheet component and no interactive representation of cloud feedbacks. The lack of an interactive ice sheet component means that sea level rise and possible decline from the melting and possibly regrowing ice sheets in reaction to the increase and decline

in radiative forcing is not included. The contribution to sea level rise from melting ice sheets is projected to become dominant in the future and reversibility of sea level rise from ice sheet melting would be delayed due to their long response time scales and possible threshold behaviour (Robinson et al., 2012; Church et al., 2013). However, there are large uncertainties for these contributions due to incomplete understanding of ice sheet dynamics (Church et al., 2013). No interactive representation of cloud feedbacks affects ocean heat uptake as investigations with AOGCMs have shown that changes in cloud cover enhance

the atmospheric cooling effect from changes in location of ocean heat uptake induced by changes in ocean circulation due to climate change (Trossman et al., 2016). Thus the model used here might underestimate the effect of changes in ocean circulation on ocean heat uptake under increasing atmospheric $CO_2$. Whether this effect between ocean heat uptake, changes in ocean circulation, and cloud cover is also in effect when the changes in ocean circulation are induced by changes in ocean mixing or how this link evolves under declining $CO_2$ concentration is unclear.

These limitations are unlikely to affect the robustness of the result that thermosteric sea level rise is not reversible on human time scales. Including the contribution from ice sheets to sea level would make sea level rise even less reversible due to the very long response timescales of ice sheet dynamics. Significant changes in the response time of thermosteric sea level changes from including dynamic cloud feedbacks are unlikely as the associated effect on ocean heat uptake is very small, only $0.01 \mathrm{Wm^{-2}}$ in a previous study (Trossman et al., 2016). Furthermore this study shows that irreversibility of GMTSL rise on human timescales

is robust against the choice of vertical diffusivity.

Generally, ocean thermal expansion induced by increased $CO_2$ concentrations does not revert to pre-industrial levels for at least a millennium after $CO_2$ concentration returned to pre-industrial levels irrespective of the choice of vertical diffusivity. Lower vertical diffusion in the ocean models, which may be closer to reality as it has been argued that ocean models are too diffusive (Hansen et al., 2011), would imply a lower GMTSL rise at first but would increase the duration of elevated sea levels after $CO_2$

concentrations are restored to pre-industrial levels. The reversibility of sea level rise would likely be prolonged further if sea





level rise from melting ice sheets would be taken into account as ice sheets respond on even longer time scales than ocean heat uptake and their contribution to sea level rise is projected to increase in the future (Church et al., 2013; Clark et al., 2016). We conclude that sea level rise from thermal expansion, and likely also from ice sheet melting, is not reversible even under strong decreases in atmospheric $CO_2$ far beyond time scales relevant to human civilization.

5 *Author contributions.* Dana Ehlert designed, executed, and analysed the model simulations and wrote the manuscript. Kirsten Zickfeld had the research idea, assisted in the data analysis and edited the manuscript.

*Acknowledgements.* K. Zickfeld acknowledges support from the Natural Sciences and Engineering Research Council of Canada (NSERC) Discovery Grant Program. This research was enabled in part by computing resources provided by Westgrid and Compute Canada.



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
