# Peer review of "Irreversible ocean thermal expansion under carbon dioxide removal"

_Earth System Dynamics, 2017_

## Referee Comment (RC1) · Anonymous Referee #1 · 30 May 2017

**Review of:** Irreversible ocean thermal expansion under negative CO$_2$ emissions

D. Ehlert and K. Zickfeld

**Overall evaluation:**

The manuscript investigates the reversibility of thermal sea-level-rise under idealized climate scenarios where CO$_2$ concentration increases exponentially, for 140 years and then are symmetrically return to pre-industrial concentration. Consistent with previous studies the manuscript finds that sea-level-fall lags the reduction in atmospheric CO$_2$ concentration and global temperature. The study shows that this behaviour is robust to the parameterization scheme used to close meridional overturning circulation.

Overall the manuscript is a solid contribution to the understanding the dynamics of the Earth system under reversibility scenarios, and fills a gap in the literature by exploring a major uncertainty left by previous studies.  I recommend that the paper undergo minor revisions.

**General Comments:**

The study is clearly intended as a sensitivity study and is generally presented in that way. However, it should be noted in the manuscript (the discussion would be a good place) that the scenarios used are highly idealized and deeply unrealistic. I accept the 1%-up 1%-down scenarios have been used extensively in the study of reversibility of climate change but the shape of these scenarios is implausible. Going from 11ppm increase in CO$_2$ to an 11ppm decrease in CO$_2$ concentration in a single year is technologically absurd. CO$_2$ concentration pathways shaped like bell-curves, would make much more sense for these kind of idealized reversibility studies.

The shape of the CO$_2$ pathway is particularly important for exploring the reversibility of sea-level-rise as the longer radiative forcing exceeds the radiative response the higher thermal sea-level rise will be and the longer it will take to dissipate the ocean heat to back to space.

To be clear I do not wish the authors to re-do their study with new CO$_2$ pathways, the study as-is is a satisfactory contribution to literature, I simply wish for the effect of the shape of the scenarios to be noted and discussed.

**Specific Comments:**

Page 2 line 6–7: This is not quite true, the scenarios used in the cited papers follow unrealistic emissions pathways (1%-up 1%-down). The actual technological feasibility of large-scale atmospheric CO$_2$ removal is unknown and will likely extend over many lifetimes.

Page 3 line 18–19: Should note that the version of the climate model you use does not have ice sheets. If I recall correctly other versions of the model do have ice sheets.

Figure 2: Why does the X-axis of b–d stop at $1^{\circ}$C?

Figure 6 & 7: Is having both panels necessary? Panel a & b appear to be identical except for the zero of the Y-axis.

Page 14 line 31: The quantity is usually called 'radiative response', not 'radiative damping'. Damping is an odd way the conceptualize the restoration of planetary energy balance.

Page 15 lines 32–35: This seems slightly confused. The ocean models are intentionally made too diffusive to close meridional overturning circulation (e.g. Munk & Wunsch, 1998) because the processes that control MOC closure occur at too small a scale to capture with most ocean models (Marshall & Speer, 2012). The lines as written seem to imply we should just turn down the diffusivity in our ocean models.

**Typos, style and grammar:**

The equations are unnumbered.

Variables in-text should be italicized.

Page 1 line 8: The abbreviation 'TSLR' is only used once.

Page 1 line 9: Change "thousand" to 'a thousand'

Page 1 line 22–24: The sentence should be re-written for clarity.

Page 1 line 24: delete 'though'

Page 1 line 25: Change 'applied' to 'tested', and 'that do' after 'scenarios'

Page 2 line 11: 'they' is ambiguous.

Page 2 line 30: "The thereby induced increase" reads wrong, please re-write for clarity.

Page 3 line 32: Spelling error in 'McWilliams'.

Page 5 line 20: Using a two character symbol for radiative forcing is confusing (it implies you are multiplying two quantities together). Maybe use $R_F$ for radiative forcing or just $F$.

Figure 1: Maybe use dashed and dotted lies to separate the various gray vertical lines.

Figure 2 caption: change 'continuous line' to 'solid line'. All the lines are continuous in the mathematical sense of the word.

Page 9 line 5: Use times symbol ($\times$ in latex) is place of '*'

**References:**

Marshall, J. and K. Speer, 2012: Closure of the meridional overturning circulation through southern ocean upwelling. Nature Geoscience, 5 (3), 171–180.

Munk, W. and C. Wunsch, 1998: Abyssal recipes ii: energetics of tidal and wind mixing. Deep Sea Research Part I: Oceanographic Research Papers, 45 (12), 1977–2010.

---

## Referee Comment (RC2) · Anonymous Referee #2 · 11 Jun 2017

The author investigated the reversibility of ocean heat uptake and sea level due to thermal expansion by linearly increasing and then decreasing atmospheric CO2 in the UVic ESCM. They found that the sea level continues for several decades after atmospheric CO2 starts to decline and does not return to the pre-industrial level. They further tested the sensitivity of vertical sub-grid scale ocean mixing and concluded that, in most cases, both rise and decline rate of sea level increase with increasing vertical ocean mixing.

This paper is potentially interesting and focuses on an important topic in the context of the Paris Agreement. I recommend this manuscript to be published in Earth System Dynamics pending on minor revisions. I have detailed my comments below.

1. The authors may want to explore more about the path of ocean heat take and

[Figure]

storage beyond the AABW formation and AMOC. For example, either observations and coupled climate models show that anthropogenic warming enters the Southern Ocean along the downwelling branch of the residual MOC (Liu et al. 2016), and heat is mostly stored preferentially stored around 45S where surface waters are subducted to the north (Armour et al. 2016). The authors' results also reveal this path, and moreover, show that it is independent from the AABW formation. In Fig. 7, either the default mixing or other settings (e.g., B&L high, constant 1.0) show a warming tongue extending from surface around 45S to deep ocean. By contrast, cooling occurs along the shelf of the Antarctic as related to the AABW formation.

2. Page 12, lines 3-4 "AMOC responses to global warming have been well studied (Stocker and Schmittner, 1997; Rahmstorf, 2006; Meehl et al., 2007) and the decline in the AMOC under warming is due to surface warming and freshening of North Atlantic surface waters (Rahmstorf, 2006)." Herein the authors may want to notice that the AMOC response to global warming is also related to AMOC stability (Liu et al. 2017). The AMOC may moderately weaken or collapse in response to $CO_2$ increase, depending on which stability regime it resides in current climate.

3. Besides the global sea level change, I (also many other readers) am interested in the regional sea level change since it is critical to particular coastal regions. I am wondering whether the authors can make some discussions about this. One key is that a great change of ocean circulation, for example, the AMOC weakening or overshoot, can modulate the dynamic sea level in the Atlantic and Southern Ocean.

References

1. Liu, W., S.-P. Xie, Z. Liu and J. Zhu, 2017, Overlooked possibility of a collapsed Atlantic Meridional Overturning Circulation in warming climate. Sci. Adv., 3, e1601666, doi:10.1126/sciadv.1601666.

2. Liu, W., S.-P. Xie and J. Lu, 2016, Tracking ocean heat uptake during the surface-warming hiatus. Nature Commun., 7, 10926, doi: 10.1038/ncomms10926.

3. Armour KC, J Marshall, J Scott, A Donohoe and ER Newsom, 2016, Southern Ocean warming delayed by circumpolar upwelling and equatorward transport. Nature Geosci., 9, 549–554.

---

## Referee Comment (RC3) · Anonymous Referee #3 · 21 Jun 2017

The authors investigated the reversibility of the ocean thermal expansion (the thermosteric sea level rise) to idealized CO2 forcing using a climate model of intermediate complexity. In their experiments, they first ramped up the CO2 concentration by 1% yearly to quadrupling and then decreased it by 1% yearly back to the preindustrial value, after which the simulations were carried on for another 1000 plus years with fixed CO2 concentration. They found that the thermosteric sea level rise is irreversible on human time scales and it continues to rise 80 years after the reversal of CO2 forcing. They further reported that the rates of sea level rise/decline in their model generally increase with higher vertical diffusivity, with exceptions of overshoot of ocean circulations.

The manuscript deals with an important issue of climate change and could contribute

to understandings of the reversibility of the climate system. The experiments and analyses are systematic and comprehensive. However, I feel that some issues should be addressed before it can be accepted. Please see my detailed comments bellow.

Major comments:

1. It would be useful if the authors can list the equilibrium climate sensitivity of the model they used, and briefly compare it with the current generation of climate models (the IPCC models). It may also help the readers if the authors can also compare the transient climate sensitivity across the sensitivity experiments with different mixing in their study. This can give the readers an idea of the effect of ocean mixing on the climate sensitivity.

2. The authors used a coupled model to carry out long simulations (longer than 1000 years). It is unclear whether the model is run with flux adjustment or the model is subject to large trend in the deep ocean. If there is any trend in the deep ocean, how does this influence the current results?

3. It surprises me that the mean sea level change lags the $CO_2$ forcing by the same amount of time (80 years) in all the sensitivity experiments. It seems that the lag time is not dependent on the details of ocean mixing. Then, the question is what is setting this lag time? Is it possible that the lag time is model dependent? It looks to me that equation on Page 5 Line 20 is the place to start discussing this problem more carefully.

4. When reviewing previous studies on the reversibility of the climate system, I think Wu et al. (2010) is worth mentioning, in which paper the author described the hysteresis behavior of the hydrological cycle in response to a ramping down of $CO_2$ forcing.

5. Mignot et al. (2007) is one of the first papers discussing the subsurface warming and the overshoot of the AMOC, which process is closely related to the overshoot of the AMOC reported in the present manuscript.

Minor comments:

1. Page 2, Line 22: Boutes et al. SHOULD BE CHANGED TO , Boutes et al.

2. Page 2, Line 26: are that is has SHOULD BE CHANGED TO are that it has

---

## Referee Comment (RC4) · Anonymous Referee #4 · 26 Jun 2017

Overview:

This manuscript explores the forward and reverse pathways of oceanic heat uptake and sea level rise and their sensitivity to sub-grid scale mixing parameterization choices in the UVic ESCM model. The experimental design is based on idealized simulations where atmospheric CO2 increases at a rate of 1%/year to quadrupling followed by a 1%/year decrease back to preindustrial values. A suite of sensitivity experiments are run based on varying a uniform constant background diffusivity, a vertically-dependent mixing scheme (Bryan & Lewis 1979), and tidal dissipation scheme (Schmittner et al. 2005). The manuscript demonstrates global sea level rise is irreversible on decadal to centennial time scales, which the authors demonstrate both in the UVic model and using a simple 2-layer diffusion model. The reversibility pathways for bottom-water

formation processes are also dependent on the nature of sub-grid scale mixing and that an increases in NADW and AABW relative to preindustrial values leads to global sea level that is below the preindustrial starting value.

Major comment:

All ocean models have some element of temperature drift. This is not discussed in the manuscript. This drift can arise for variety of reasons, including spurious mixing in depth-based coordinate models. This drift can be on the order of several tenths of a degree per century for the global volume average temperature. Drifts of this magnitude are non-negligible on the millennial timescales discussed in this manuscript. Furthermore, the magnitude of this drift is directly linked to the strength of the sub-grid scale mixing parameterizations used in ocean models. After the 6,000 year spin up of the model, suddenly varying the mixing coefficients in the model will produce noticeable changes in global ocean temperature and sea level regardless of changes in atmospheric CO2. It would be preferable to run control simulations for each of the mixing perturbations considered in this study. The difference plots in each of the figures should be relative to their respective control simulations rather than year 0 of the simulation. This approach would more cleanly separate the response to forcing from the inherent model drift.

Minor Comments:

Page 1, Line 22: Recovery on what timescale? Page 2, Line 2: How do the negative emissions in RCP2.6 compare with decreasing atmospheric CO2 at a rate of 1%/year? Page 2, Line 8: Sea ice decline should be sea ice recovery Page 3, Line 6: This is an excellent place to discuss the known constraints that the modeling community does have in regards to sub-grid scale mixing? (i.e. how big is the uncertainty?) Page 5, Line 17: On what timescale? Page 6, Line 1: Does Delta-T imply the near-surface air temperature? Page 12, Line 1: What are precise definitions used to assess AMOC and AABW rates? e.g. Is it the annual time series of the meridional overturning at the

latitude of the RAPID array (26.5N)?

Figure Comments:

Figure 1, panels b & d: A line denoting zero would be helpful. Figure 2, panels c & d: What are the surface values? They could potentially be very unrealistic.

———————————————————

---

## Author Response (AR1)

**Referee1**

**Review of:** Irreversible ocean thermal expansion under negative CO2 emissions D. Ehlert and K. Zickfeld **Overall evaluation:**

The manuscript investigates the reversibility of thermal sea-level-rise under idealized climate scenarios where CO2 concentration increases exponentially, for 140 years and then are symmetrically return to pre-industrial concentration. Consistent with previous studies the manuscript finds that sea-level-fall lags the reduction in atmospheric CO2 concentration and global temperature. The study shows that this behaviour is robust to the parameterization scheme used to close meridional overturning circulation.

Overall the manuscript is a solid contribution to the understanding the dynamics of the Earth system under reversibility scenarios, and fills a gap in the literature by exploring a major uncertainty left by previous studies. I recommend that the paper undergo minor revisions.

Response: We want to thank the reviewer for a helpful and detailed review.

**General Comments:**

The study is clearly intended as a sensitivity study and is generally presented in that way. However, it should be noted in the manuscript (the discussion would be a good place) that the scenarios used are highly idealized and deeply unrealistic. I accept the 1%-up 1%-down scenarios have been used extensively in the study of reversibility of climate change but the shape of these scenarios is implausible. Going from 11ppm increase in CO2 to an 11ppm decrease in CO2 concentration in a single year is technologically absurd. CO2 concentration pathways shaped like bell-curves, would make much more sense for these kind of idealized reversibility studies.

The shape of the CO2 pathway is particularly important for exploring the reversibility of sealevel-rise as the longer radiative forcing exceeds the radiative response the higher thermal sealevel rise will be and the longer it will take to dissipate the ocean heat to back to space.

To be clear I do not wish the authors to re-do their study with new CO2 pathways, the study as-is is a satisfactory contribution to literature, I simply wish for the effect of the shape of the scenarios to be noted and discussed.

Response: Thank you for pointing out this gap in the discussion. We agree and have included a discussion of it in the manuscript. Adjusted text (Page 14 Line 31 -Page 15 Line 1): The net negative CO2 emissions required to achieve a 1% yearly decline in atmospheric CO2 are likely unfeasible with current technologies (Boucher et al., 2012; Tokarska and Zickfeld, 2015) and the results shown in this study are therefore conservative. Thus, also the strong change between positive and negative emissions needed to achieve the turn between a 1% yearly CO2 increase and a decrease is technologically unlikely.

**Specific Comments:**

Page 2 line 6 - 7: This is not quite true, the scenarios used in the cited papers follow unrealistic emissions pathways (1%-up 1%-down). The actual technological feasibility of large-scale atmospheric CO2 removal is unknown and will likely extend over many lifetimes. Response: This is not true for the Tokarska and Zickfeld, 2015 paper, where emission reduction rate were restricted to a maximum decline of 4% per year relative to year 2000.

Page 3 line 18 - 19: Should note that the version of the climate model you use does not have ice sheets. If I recall correctly other versions of the model do have ice sheets. Response: Thank you for pointing out this negligence (Page 3 Line 23).

Figure 2: Why does the X-axis of b - d stop at 1oC?

Response: To enable readability of temperature differences in the deep ocean between model versions. The surface ocean temperatures can be inferred from global mean surface air temperature time series.

Figure 6 & 7: Is having both panels necessary? Panel a & b appear to be identical except for the zero of the Y-axis.

Response: We prefer keeping both panels, as the b panels are essential the show the stronger intensification/overshoot of some of the low mixing cases relative to the high mixing cases. However, only showing the change relative to year 0 might be misleading. We chose to combine Figure 6 and 7 into one figure.

Page 14 line 31: The quantity is usually called 'radiative response', not 'radiative damping'. Damping is an odd way the conceptualize the restoration of planetary energy balance.

Response: We choose to use the same wording as in Zickfeld et.al. 2017 to clarify to readers that the same underlying theoretical idea is used.

Page 15 lines 32 – 35: This seems slightly confused. The ocean models are intentionally made too diffusive to close meridional overturning circulation (e.g. Munk & Wunsch, 1998) because the processes that control MOC closure occur at too small a scale to capture with most ocean models (Marshall & Speer, 2012). The lines as written seem to imply we should just turn down the diffusivity in our ocean models.

Response: Thank you for pointing out this potential misunderstanding. We rephrased the sentences to make this point more clear (Page 16 Line 11-16).

**Typos, style and grammar:**

The equations are unnumbered. Response: We included numbering for the equations.

Variables in-text should be italicized. Response: They have been italicized now.

Page 1 line 8: The abbreviation 'TSLR' is only used once. Response: We now also use the abbreviation GMTSL (global mean thermosteric sea level) in the abstract and deleted the abbreviation TSLR.

Page 1 line 9: Change "thousand" to 'a thousand' Response: Has been included (Page 1 Line 9).

Page 1 line 22 - 24: The sentence should be re-written for clarity. Response: The sentence has been re-phrased to improve clarity (Page 1 Line 24-26).

Page 1 line 24: delete 'though' Response: Has been deleted.

Page 1 line 25: Change 'applied' to 'tested', and 'that do' after 'scenarios' Response: Changed accordingly.

Page 2 line 11: 'they' is ambiguous. Response: The sentence has been adjusted to improve clarity (Page 2 Line 13).

Page 2 line 30: "The thereby induced increase" reads wrong, please re-write for clarity. Response: The sentence has been rephrased (Page 2 Line 34 -Page 3 Line 2).

Page 3 line 32: Spelling error in 'McWilliams'. Response: It has been changed (Page 4 Line 5).

Page 5 line 20: Using a two character symbol for radiative forcing is confusing (it implies you are multiplying two quantities together). Maybe use RF for radiative forcing or just F. Response: RF has been replaced with RF in the text and equations.

Figure 1: Maybe use dashed and dotted lies to separate the various gray vertical lines. Response: The line styles have been changed.

Figure 2 caption: change 'continuous line' to 'solid line'. All the lines are continuous in the mathematical sense of the word.

Response: Thank you for pointing to this error. It has been changed accordingly.

Page 9 line 5: Use times symbol (\$\times\$ in latex) is place of '\*' Response: It has been replaced (Page10 Line 9-14).

**References:**

Marshall, J. and K. Speer, 2012: Closure of the meridional overturning circulation through southern ocean upwelling. Nature Geoscience, 5 (3), 171 - 180.

Munk, W. and C. Wunsch, 1998: Abyssal recipes ii: energetics of tidal and wind mixing. Deep Sea Research Part I: Oceanographic Research Papers, 45 (12), 1977 - 2010.

Earth Syst. Dynam. Discuss., https://doi.org/10.5194/esd-2017-45-RC2, 2017 © Author(s) 2017. This work is distributed under the Creative Commons Attribution 3.0 License.

The author investigated the reversibility of ocean heat uptake and sea level due to thermal expansion by linearly increasing and then decreasing atmospheric CO2 in the UVic ESCM. They found that the sea level continues for several decades after atmo- spheric CO2 starts to decline and does not return to the pre-industrial level. They fur- ther tested the sensitivity of vertical sub-grid scale ocean mixing and concluded that, in most cases, both rise and decline rate of sea level increase with increasing vertical ocean mixing.

This paper is potentially interesting and focuses on an important topic in the context of the Paris Agreement. I recommend this manuscript to be published in Earth System Dynamics pending on minor revisions. I have detailed my comments below.

Response: We want to thank the reviewer for a positive review.

1. The authors may want to explore more about the path of ocean heat take and storage beyond the AABW formation and AMOC. For example, either observations and coupled climate models show that anthropogenic warming enters the Southern Ocean along the downwelling branch of the residual MOC (Liu et al. 2016), and heat is mostly stored preferentially stored around 45S where surface waters are subducted to the north (Armour et al. 2016). The authors' results also reveal this path, and moreover, show that it is independent from the AABW formation. In Fig. 7, either the default mixing or other settings (e.g., B&L high, constant 1.0) show a warming tongue extending from surface around 45S to deep ocean. By contrast, cooling occurs along the shelf of the Antarctic as related to the AABW formation.

Response: Thank you for pointing to these interesting articles. We included a discussion on the warming tongue seen in Figure 3 and the findings from the above mentioned articles in the manuscript. Added text (Page14 Line18-22):

A warming tongue just north of the Antarctic Circumpolar Current can be observed for most model version even towards the end of the simulations (Figure 3). This agrees with observations and simulations, which show strong ocean heat uptake of anthropogenic induced warming in the Southern Ocean along the downwelling branch of the residual meridional overturning circulation (Liu et al., 2017). The majority of this heat is stored around 45°S where surface waters sink down and travel northward (Armour et al., 2016).

2.Page 12, lines 3-4 "AMOC responses to global warming have been well studied (Stocker and Schmittner, 1997; Rahmstorf, 2006; Meehl et al., 2007) and the decline in the AMOC under warming is due to surface warming and freshening of North Atlantic surface waters (Rahmstorf, 2006)." Herein the authors may want to notice that the AMOC response to global warming is also related to AMOC stability (Liu et al. 2017). The AMOC may moderately weaken or collapse in response to CO2 increase, depending on which stability regime it resides in current climate.

Response: Thank you for pointing to this interesting article we cited in the manuscript (Page 13 Line 9).

3. Besides the global sea level change, I (also many other readers) am interested in the

regional sea level change since it is critical to particular coastal regions. I am wondering whether the authors can make some discussions about this. One key is that a great change of ocean circulation, for example, the AMOC weakening or overshoot, can modulate the dynamic sea level in the Atlantic and Southern Ocean.

Response: Regional sea level changes in an interesting research question. However, this manuscript focusses on global mean sea level rise and including analyses and discussions around regional sea level changes is beyond the scope of this paper.

**References**

1. Liu, W., S.-P. Xie, Z. Liu and J. Zhu, 2017, Overlooked possibility of a collapsed Atlantic Meridional Overturning Circulation in warming climate. Sci. Adv., 3, e1601666, doi:10.1126/sciadv.1601666.

2. Liu, W., S.-P. Xie and J. Lu, 2016, Tracking ocean heat uptake during the surfacewarming hiatus. Nature Commun., 7, 10926, doi: 10.1038/ncomms10926.

3. Armour KC, J Marshall, J Scott, A Donohoe and ER Newsom, 2016, Southern Ocean warming delayed by circumpolar upwelling and equatorward transport. Nature Geosci., 9, 549–554.

The authors investigated the reversibility of the ocean thermal expansion (the ther- mosteric sea level rise) to idealized CO2 forcing using a climate model of intermedi- ate complexity. In their experiments, they first ramped up the CO2 concentration by 1% yearly to quadrupling and then decreased it by 1% yearly back to the preindustrial value, after which the simulations were carried on for another 1000 plus years with fixed CO2 concentration. They found that the thermosteric sea level rise is irreversible on human time scales and it continues to rise 80 years after the reversal of CO2 forcing. They further reported that the rates of sea level rise/decline in their model generally increase with higher vertical diffusivity, with exceptions of overshoot of ocean circulations.

The manuscript deals with an important issue of climate change and could contribute to understandings of the reversibility of the climate system. The experiments and analyses are systematic and comprehensive. However, I feel that some issues should be addressed before it can be accepted. Please see my detailed comments bellow.

Response: We want to thank the reviewer for a helpful and positive review.

Major comments:

1. It would be useful if the authors can list the equilibrium climate sensitivity of the model they used, and briefly compare it with the current generation of climate models (the IPCC models). It may also help the readers if the authors can also compare the transient climate sensitivity across the sensitivity experiments with different mixing in their study. This can give the readers an idea of the effect of ocean mixing on the climate sensitivity. Response: We included the equilibrium climate sensitivity of the UVic model. We also included a list of the transient climate sensitivity of all model versions and mention the IPCC AR5 and EMIC ranges. Added text (Page 4 Line 24 – Page 5 Line 7): The Transient Climate Response (TCR), which is the GMSAT at a doubling of atmospheric CO2 relative to pre-industrial, for the different model versions has a range of 1.58 to 2.24 °C (Table 1). This range lies within the EMIC TCR range of 0.8 to 2.5 °C (Eby et al., 2013). The Equilibrium Climate Sensitivity (ECS), which is the equilibrium temperature change at doubling of pre-industrial CO2 concentration, for the default model version is 3.5 °C (Eby et al., 2013). For comparison, the likely ranges for the TCR and ECS in the last Intergovernmental Panel for Climate Change (IPCC) report are 1 to 2.5 °C and 1.5 to 4.5 °C, respectively (Collins et al., 2013). A discussion of the ocean heat uptake efficiency between the model versions and comparison to other models can be found in Ehlert et al. (2017).

2. The authors used a coupled model to carry out long simulations (longer than 1000 years). It is unclear whether the model is run with flux adjustment or the model is subject to large trend in the deep ocean. If there is any trend in the deep ocean, how does this influence the current results?

Response: The model does not require flux adjustments. There is a small drift in the deep ocean temperature but it is negligible for most model versions. For the model versions where the drift in the deep ocean gets close to the magnitude of the observed temperature change relative to year 0, taking the drift into account does not affect the results. Drift is now mentioned in the Section Simulations (Page 4 Line 20-24) and a detailed discussion is

added as supplementary material.

3. It surprises me that the mean sea level change lags the CO2 forcing by the same amount of time (80 years) in all the sensitivity experiments. It seems that the lag time is not dependent on the details of ocean mixing. Then, the question is what is setting this lag time? Is it possible that the lag time is model dependent? It looks to me that equation on Page 5 Line 20 is the place to start discussing this problem more carefully.

Response: Thank you for bringing this point to our attention. It seems the text might be a somewhat unclear. The lag is not always exactly 80 years for all model versions but ranges between 67 to 86 years. However, the lag is always on multi-decadal time scales. We adjusted the manuscript accordingly (Page 9 Line 5).

4. When reviewing previous studies on the reversibility of the climate system, I think Wu et al. (2010) is worth mentioning, in which paper the author described the hysteresis behavior of the hydrological cycle in response to a ramping down of CO2 forcing. Response: Thank you for pointing to this interesting study, we mentioned it in our manuscript (Page 2 Line 9-10).

5. Mignot et al. (2007) is one of the first papers discussing the subsurface warming and the overshoot of the AMOC, which process is closely related to the overshoot of the AMOC reported in the present manuscript.

Response: Thank you for pointing to this interesting article. We do not believe that the mechanism for the overshoot discussed in the Mignot paper and in our manuscript are the same. In the Mignot paper the reason for the overshoot is a subsurface warming induced by a complete shutdown of deep ventilation in the North Atlantic. In our case the AMOC does not shutdown at any point in the simulation and thus deep ventilation in the North Atlantic never shuts down completely either.

Minor comments:

- Page 2, Line 22: Boutes et al. SHOULD BE CHANGED TO, Boutes et al. Response: We assume this points to the usage of parenthesis. This is not a mistake. We refer to the initial description of the 2-layer model by Gregory 2000 and Geoffroy et.al. 2013 but then describe how Bouttes et.al. 2013 used it under declining radiative forcing. We rearranged the sentence to improve clarity (Page 2 Line 24-25).
- 2. Page 2, Line 26: are that is has SHOULD BE CHANGED TO are that it has Response: This has been edited in the text. Thank you for pointing out this typo (Page 2 Line 29).

This manuscript explores the forward and reverse pathways of oceanic heat uptake and sea level rise and their sensitivity to sub-grid scale mixing parameterization choices in the UVic ESCM model. The experimental design is based on idealized simulations where atmospheric CO2 increases at a rate of 1%/year to quadrupling followed by a 1%/year decrease back to preindustrial values. A suite of sensitivity experiments are run based on varying a uniform constant background diffusivity, a vertically-dependent mixing scheme (Bryan & Lewis 1979), and tidal dissipation scheme (Schmittner et al. 2005). The manuscript demonstrates global sea level rise is irreversible on decadal to centennial time scales, which the authors demonstrate both in the UVic model and using a simple 2-layer diffusion model. The reversibility pathways for bottom-water formation processes are also dependent on the nature of sub-grid scale mixing and that an increases in NADW and AABW relative to preindustrial values leads to global sea level that is below the preindustrial starting value.

Response: We want to thank the reviewer for the helpful and positive review.

**Major comment:**

All ocean models have some element of temperature drift. This is not discussed in the manuscript. This drift can arise for variety of reasons, including spurious mixing in depthbased coordinate models. This drift can be on the order of several tenths of a degree per century for the global volume average temperature. Drifts of this magnitude are nonnegligible on the millennial timescales discussed in this manuscript. Furthermore, the magnitude of this drift is directly linked to the strength of the sub-grid scale mixing parameterizations used in ocean models. After the 6,000 year spin up of the model, suddenly varying the mixing coefficients in the model will produce noticeable changes in global ocean temperature and sea level regardless of changes in atmospheric CO2. It would be preferable to run control simulations for each of the mixing perturbations considered in this study. The difference plots in each of the figures should be relative to their respective control simulations rather than year 0 of the simulation. This approach would more cleanly separate the response to forcing from the inherent model drift.

Response: Each model version with the changed mixing is spun up to account for the effect of the changes in mixing as described in the simulation section. Thus changes relative to year 0 are changes relative to the end of spin up state for each individual mixing parameter scheme/setting. Therefore, the changes in global mean temperature and sea level account only for changes induced by the forcing and not induced by changes in mixing. The effect of mixing can then be inferred by comparing the different lines in 1d plots or subplots in Figure 3.

The drift of global mean ocean temperature is negligible after 6000 year spin up. The drift of deep ocean temperature is small as well but get close to the temperature change in the deep ocean for the forced simulations. This, however, does not affect the findings discussed in this study. Drift is now mentioned in the Section Simulations (Page 4 Line 20-24) and discussed in detail in the supplementary material.

Minor Comments:

Page 1, Line 22: Recovery on what timescale? Response: On multi-millennial time scales. We included this here (Page 1 Line 22-23).

Page 2, Line 2: How do the negative emissions in RCP2.6 compare with decreasing atmospheric CO2 at a rate of 1%/year?

Response: We included the highest rates for negative emissions (around 20 PgC/yr for 1% yearly atmospheric CO2 decline and 1.6PgC/yr for RCP2.6) in the discussion section, where the limitations of the 1% scenario are discussed (Page 15 Line 2-3).

Page 2, Line 8: Sea ice decline should be sea ice recovery Response: We added recovery into the sentence (Page 2 Line 9).

Page 3, Line 6: This is an excellent place to discuss the known constraints that the modeling community does have in regards to sub-grid scale mixing? (i.e. how big is the uncertainty?) Response: It is difficult to generalize uncertainties for the different diffusivities and going into detail about the different diffusivities at this point in the manuscript does not seem adequate as they have not been introduced specifically. Furthermore, the uncertainties for the parameters are also model specific and mixing parameter values for varying ESM are very difficult to access. As discussed in the simulation section, the range for the background diffusivity in the tidal mixing scheme is based on uncertainty studies. There are no uncertainty studies for Uvic for the other vertical mixing schemes to our knowledge.

Page 5, Line 17: On what timescale?

Response: GMTSL change is proportional to GMSAT changes on multi-centennial time scales. Thank you for pointing to this missing information. We included it now at this point (Page 7. Line 9-10).

Page 6, Line 1: Does Delta-T imply the near-surface air temperature? Response: In the equation  $\Delta T$  reflects the temperature change in the 0-D EBM. When this approach is being applied to the model out put as in Figure 4, yes change in GMSAT is used as  $\Delta T$ . This is being described in the caption of Figure 3.

Page 12, Line 1: What are precise definitions used to assess AMOC and AABW rates? e.g. Is it the annual time series of the meridional overturning at the latitude of the RAPID array (26.5N)?

Response: Both AMOC and MOC in the Southern Ocean are now computed from maximum respectively minimum of the zonally averaged stream function below a depth of 400 m. The stream function is calculated from decadal averages of the velocity. The maximum of the zonally averaged stream function occurs between 30N and 40N and the minimum occurs between 0S and 70S. For consistency we now use decadal averages for the AMOC calculations as well. We expand the information given on the calculations in caption of Figures 6 (Figures 6 and 7 are now combined into Figure 6).

Figure Comments:

Figure 1, panels b & d: A line denoting zero would be helpful.

Response: We included those lines now.

Figure 2, panels c & d: What are the surface values? They could potentially be very unrealistic.

Response: The surface values are cropped out here as otherwise the differences in temperature change in deeper layers between the model version would not be visible. The surface values are very similar to the global mean surface air temperatures for those points

in time, which can be inferred from the time evolution plots of the global mean surface air temperatures (Figure 1).

---

## Author Response (AR2)

Referee#2:

Line 21 in Page 14, "Liu et al., 2017" should be "Liu et al., 2016"
Liu, W., S.-P. Xie and J. Lu, 2016, Tracking ocean heat uptake during the surface- warming hiatus. Nature Commun., 7, 10926

There is no discussion about Southern Ocean in Liu et al. (2017).

Response: Thank you for pointing out this mistake. We have changed now cite Liu et al. (2016) on page 14, line 21.

[revised manuscript text omitted]